# Methadone Maintenance and QT-Interval: Prevalence and Risk Factors—Is It Effective to Switch Therapy to Levomethadone?

**DOI:** 10.3390/biomedicines11082109

**Published:** 2023-07-26

**Authors:** Laura Santin, Giuseppe Verlato, Ahmad Tfaily, Roberto Manera, Giuseppe Zinfollino, Francesca Fusina, Fabio Lugoboni

**Affiliations:** 1Addiction Unit, Department of Medicine, Verona University Hospital, 37134 Verona, Italy; laura.santin@univr.it; 2Section of Epidemiology and Medical Statistics, Department of Diagnostics and Public Health, University of Verona, 37134 Verona, Italy; giuseppe.verlato@univr.it (G.V.); tfailya@gmail.com (A.T.); 3Addiction Department, ULSS 2, 31100 Treviso, Italy; roberto.manera@aulss2.veneto.it (R.M.); giuseppe.zinfollino@aulss2.veneto.it (G.Z.); 4Padova Neuroscience Center, University of Padua, 35122 Padua, Italy; francescafusina@gmail.com

**Keywords:** methadone, levomethadone, QTc, prolongation, treatment, heroin, therapy-switch

## Abstract

Methadone is a chiral synthetic opioid primarily used to treat heroin and prescription-opioid addiction: the (R)-enantiomer (Levomethadone) activates the µ-opioid receptor more potently than the (S)-enantiomer, which is a more potent blocker of the hERG potassium channels, resulting in QTc prolongation. The purpose of this retrospective study was to assess the effect of methadone on the QTc interval and to investigate the benefits of Levomethadone. The electrocardiograms of 165 patients taking methadone at various dosages and for different periods of time were examined: the QTc value was manually measured and then adjusted using Bazett’s formula. Data analysis revealed a linear association between the dosage of methadone and QTc length; no correlation was found between the QTc value and gender, age, or duration of therapy. In total, 14% of the sample (23 patients) showed a prolongation of the QTc interval (>470 ms in males and >480 ms in females); 10 of the 23 patients with QTc elongation underwent a change of therapy from Methadone to Levomethadone—in 90% of these patients, a normalization in the QTc length was established. This study confirmed the role of methadone, specifically its dosage, in QTc prolongation and the efficiency of Levomethadone as an adequate therapeutic substitute in these circumstances. This study validates the importance of careful electrocardiographic monitoring in methadone-treated patients.

## 1. Introduction

Methadone (MTD) is the most commonly used drug in the world for treating heroin and prescription-opioid addiction due to its safety, efficacy, and low cost [1]. Furthermore, due to its distinctive pharmacokinetic and pharmacodynamic properties—such as its high oral bioavailability, its longer half-life when compared to other opioids, its non-competitive antagonist action on NMDA receptors, and its inhibitory action on serotonin and noradrenaline reuptake—MTD is a drug that can be used to treat both chronic and acute pain [2]; MTD is now used to treat both acute post-operatory pain and chronic pain, both cancer and non-cancer related [3,4,5]. It is estimated that over 300,000 patients in Europe are receiving MTD maintenance therapy for opioid addiction (750,000 patients worldwide); MTD was added to the WHO’s list of essential drugs in 2005 due to the reduction in mortality, relapses, and illicit activities it ensures [6]. The benefits of using MTD as a treatment for opioid-addicted patients include cravings control, the disappearance of the effect–abstinence cycle, the complete remission of withdrawal symptoms, and the tolerance to MTD’s euphoric, sedative, and narcotic effects [1]. 

MTD is a chiral compound and its two enantiomers (R and S) have different pharmacokinetic and pharmacodynamic properties; it is typically administered orally, with the plasmatic peak occurring after 2–4 h [6]. It has a half-life of 24 to 36 h, allowing for daily administration [6]. MTD and its active and inactive metabolites are excreted by the kidney after being metabolized in the liver by cytochrome P450—specifically, cytochrome CYP3A4, CYP2B6, and CYP2D6 [6]. The two enantiomers have similar bioavailability, but LevoMTD has a higher distribution volume and a longer half-life [6]. 

MTD is an agonist of µ-opioid receptors, with the R-enantiomer (Levomethadone) being a stronger agonist than the S-enantiomer; levomethadone (LevoMTD), indeed, has a tenfold higher affinity for µ1 and µ2 receptors and an analgesic potential that is 8 to 50 times greater [7]. Constipation, nausea, sleepiness, vomiting, headache, and erectile dysfunction are the most common side effects that can occur during MTD treatment; these effects usually appear during the induction phase and generally disappear after a few weeks of treatment—the only exception being constipation [1]. Despite its widespread use and relative safety and handling, MTD use—particularly when long-term—is not without risk; the risks include intoxication and overdose, respiratory depression, and sudden cardiac death due to torsade de points (TdP) [8]. TdP is a polymorphic ventricular tachycardia that occurs in patients with an elongated QT interval; this event occurs in 5% of antiarrhythmic drug patients and less frequently in patients on other drugs such as MTD [8,9,10]. These risks are primarily related to the drug’s long half-life, high bioavailability, and tendency to accumulate in the organism due to slow excretion [8].

In the general population, the elongation of the ECG’s QT interval can be caused by genetic variants or by acquired long QT syndrome—the main risk factor for which is drug use [11]. The QT interval reflects the duration of the ventricular depolarization and repolarization caused by inward depolarizing currents (involving sodium and calcium channels) and outward repolarizing currents (involving potassium channels) [11]. A malfunctioning in these channels can lead to an excess of intracellular positive ions, prolonging the ventricular repolarization and the QT interval [11]. A variety of factors can cause the QT interval to be prolonged; those associated with a very strong level of evidence are the use of drugs with a known risk of QT interval prolongation, the use of cardiovascular drugs (such as diuretics and antiarrhythmics) and hypokalemia. Age (>65 years), female sex, smoking, ischemic cardiomyopathy, hypertension, arrhythmia, thyroid disfunction, hypocalcemia and the use of drugs with a possible or conditional risk for QT prolongation are factors with a strong level of evidence; Body Mass Index (BMI), basal prolongation of the QT interval, cardiovascular drugs, septic shock, liver failure, the Charlson Comorbidity Index, hypochloremia and hyponatremia are factors with a moderate level of evidence; hyperlipidemia, neurologic disorders, diabetes, kidney failure and depression only have a low level of evidence of association with QT prolongation [12].

The list of drugs that can prolong the QT interval and cause TdP is constantly updated and can be found online (www.crediblemeds.org), where drugs are classified as having a known, possible, or conditional risk of TdP (Table 1) [13].

MTD use can result in cardiologic side effects—most notably a prolongation of the ECG’s QT interval; this can happen with a wide range of dosages, including those recommended for maintenance therapy [10,14,15]. MTD’s action on the cardiac hERG (human ether-a-go-go-related gene) channels, which are voltage-dependent potassium channels responsible for the cardiac repolarizing current, is the mechanism by which it can prolong the QT interval [15]. MTD, in fact—like the majority of other QT-prolonging drugs—acts by blocking the hERG potassium channels involved in ventricular repolarizing currents [16]. As MTD binds to the channel, it inhibits the delayed rectifier potassium current—slowing down the cardiac repolarization (caused by the outward flux of potassium ions); the resulting excess of positive ions inside the cells prolongs the cardiac ventricular repolarization, resulting in an increase in the QT interval seen on the ECG [17]. The two MTD enantiomers, R and S, inhibit hERG channels in different ways: R-MTD inhibits 40% of the channels while S-MTD inhibits 65% of them. Furthermore, R-MTD inhibits the channels much less potently than the S enantiomer [18,19]. As R-MTD—or LevoMTD—is the main channel responsible for the drug’s therapeutic activity, and as it has a 3.5-fold lower affinity for hERG channels than S-MTD, using LevoMTD alone could be beneficial in order to reduce the risk of TdP without sacrificing the racemic mixture’s therapeutic efficiency [6].

The purpose of this study was to determine the prevalence of QT prolongation in the studied population and to assess the other factors that influence the QT interval. Another goal was to compare manual and automatic QT interval readings and to assess the benefits of using LevoMTD in patients with prolonged QT intervals.

## 2. Materials and Methods

This retrospective study was carried out in three Addiction Services (Servizi Dipendenze; Ser.D) of the Social Health Unit of Treviso, North-Eastern Italy: data were collected in the Ser.Ds of Treviso, Oderzo, and Castelfranco Veneto for a total of 165 patients (31 women and 134 men) on Methadone (Molteni Farmaceutici S.p.A., Scandicci, Firenze, Italy) maintenance therapy at various dosages. Gender, age, length of MTD treatment, daily MTD dosage, and other drugs taken by the patients at the time of the ECG were the variables considered for each patient. Furthermore, the control ECG was evaluated in patients who had a prolonged QT interval during MTD treatment and were then switched to LevoMTD (Ellepalmiron®, Molteni Farmaceutici S.p.A., Scandicci, Firenze, Italy).

The QT interval was measured manually, as many studies have shown that manual measurements are more accurate than automatic measurements and that the latter underestimate the QT interval [20,21,22]. The QT interval begins with the onset of the QRS complex, the first manifestation of ventricular depolarization, and ends with the termination of the T wave, the final manifestation of ventricular repolarization [23]. The correct evaluation of the QT interval requires the selection of the appropriate cardiac lead; in general, lead II or V5 are preferable, even though the American Heart Association recommends evaluating the lead with the best visibility of the T wave [24]. When suitable, the II and V5 leads were evaluated in this study; otherwise, the better-defined one was used. After selecting the best lead, the QT interval was measured using the tangent technique to determine the end of the T wave [25]. As the QT interval varies with heart rate, it is necessary to use a correction formula after measuring the QT interval to obtain the QTc value (corrected QT interval), which is related to the risk of TdP [26]. Bazett’s formula is the most commonly used correction formula, and it was also used in this study [26]:(1)$$QTc=\frac{QT}{\sqrt{RR}}$$

RR indicates the RR interval, measured in seconds, preceding the QT under consideration, measured in milliseconds [26]. Although Bazett’s formula appears to be inaccurate for extreme values of heart rate and many alternative formulas are available (such as Bogossian’s formula for values of QRS greater than 120 ms or Fridericia’s formula for bradycardiac patients), it has never been proven that these formulas are superior to Bazett’s [27].

Qualitative variables were reported as absolute frequencies and percentages, quantitative variables as mean and standard deviation (SD) if symmetrically distributed, or as median and interquartile range if asymmetrically distributed. As there is no gold standard, the concordance between manually and automatically measured QTc was assessed using the Bland–Altman method. The assumption of normality was verified by the Shapiro–Wilk and Shapiro–Francia tests. The assumption of homoskedasticity was tested by the variance ratio test when comparing two sample means or by the Breusch–Pagan/Cook–Weisberg test for heteroskedasticity when performing linear regression. As both assumptions were met for QTc, measured either manually or automatically, parametric statistics was employed.

The *t* test was used to determine whether there was a significant QTc difference between genders. Simple linear regression was used to assess the relationship between QTc interval on one hand and age, duration, and dosage of MTD treatment on the other. The influence of methadone treatment (daily dosage and duration) on the QTc interval was further investigated by a multiple linear regression model—controlling for gender, age, and other treatments (antidepressant, typical, and atypical antipsychotic drugs). To evaluate changes in QTc interval after switching to levoMTD, we used the paired *t*-test.

Statistical analyses were performed using STATA software, version 16.1 (StataCorp LLC, College Station, TX, USA).

## 3. Results

### 3.1. Sample Characteristics

The total number of evaluated patients was 165, with 31 women (18.8%) and 134 men (81.2%). The mean (SD) age was 42 (11.1) years, with a range from 18 to 68 years.

The median duration of MTD treatment was 10 years (p25–p75 = 3–22 years: range = 0.25–41 years); the median daily dosage of MTD was 80 mg (p25–p75 = 40–120 mg), with a minimum of 5 mg and a maximum of 440 mg. Both distributions showed significant positive asymmetry.

In addition to MTD, 35% of the population (57 patients) used benzodiazepines, 18.8% (31 patients) used antidepressants, 15.2% (25 patients) used atypical antipsychotics, 6% used typical antipsychotics, and 4.8% used mood stabilizers.

### 3.2. ECGs Evaluation

As the ECGs were being performed, all patients had a sinusal rhythm. The mean (SD) heart rate (HR) was 71 (12) bpm, and the mean (SD) QRS length was 95 (11) ms.

When the manually measured QTc values were compared to the automatic ones, no significant difference was found; on the contrary, the two measurements had a nearly perfect correspondence, as shown both by the scatterplot (Figure 1) and the Bland–Altman plot (Figure 2). Anyway, because many articles claim that manual measurement is superior, we used manually measured QTc values for the statistical analysis [20,21].

The mean (SD) QTc value (manually measured) in the sample was 431 (±32) ms (range 356–528 ms). The QTc did not significantly differ between gender, amounting to 435 ± 30 ms in women and 430 ± 33 ms in men (*p* = 0.489; Figure 3). The QTc interval was not affected by age (Figure 4).

### 3.3. Correlations with Methadone Treatment

QTc duration was positively related to MTD daily dosage; the QTc interval increased on average by 11.3 ms when the MTD daily dose increased by 100 mg (*p* = 0.004; Figure 5). In addition, the association between MTD daily dose—coded over four levels—and QTc interval is presented in Figure 6. On the other hand, no correlation was recorded between the QTc interval and duration of MTD treatment (Figure 7).

The daily dose of MTD emerged as the only independent risk factor for QTc prolongation in multivariate analyses; the QTc increased by 28 msec when the dose of MTD increased from <50 to 100–149 mg/day.

On the other hand, the duration of MTD therapy, gender, age, use of antidepressants, and use of typical or atypical antipsychotic drugs were not associated with QTc prolongation (Table 2).

### 3.4. Patients with QTc Prolongation

Using 470 ms for men and 480 ms for women as cut-offs [28], 23 patients (14%) with QTc elongation were identified: there were three women and twenty men among these patients (Table 3).

When we examined if these patients were taking medications associated with QTc prolongation, we discovered that none of them were taking drugs with a known risk of TdP. However, three of the 23 patients (13%) were taking medications with a possible risk of TdP (atomoxetine, clotiapine, paliperidone, and ritonavir), and two (8,7%) were taking medications with a conditional risk of TdP (trazodone). The other 18 patients (78,3%) were not taking any medication.

### 3.5. Changes to levoMTD Therapy

Among the 23 patients with prolonged QTc intervals, 10 (nine men and one woman) switched from MTD to LevoMTD maintenance therapy according to the medication’s instruction diagram, which states that levomethadone is roughly two times more active than methadone and should therefore be prescribed at half the patients’ previous dose of methadone [29]. Following the therapy change, the QTc values were reduced in nine of these patients (Table 4, Figure 8).

At the time of the control ECG, the mean duration of Levomethadone treatment was 19.8 months, with a minimum of 1.6 months and a maximum of 48 months. The mean LevoMTD dosage was 70.5 mg/day.

Three of these ten patients were using medications that carried a possible and conditional risk of lengthening the QTc; however, they were taking these medications while receiving both methadone and levomethadone treatment—thus that variable remained constant before and after the therapy switch.

The mean (SD) decrease in QTc after switching to levoMTD amounted to 47 ms (29). Only one subject had an increase in QTc after switching therapy, and this may be due to inter-subject variability in drug responses. It is worth noting that the only patient who did not have a normalization of the QTc interval following the switch in therapy was the patient who had been on levoMTD for the shortest amount of time; the control ECG was performed only 1 month and 20 days after the start of levoMTD therapy.

## 4. Discussion

This study found that more than one-eighth of MTD-treated patients had a QTc prolongation. This value is not significantly different from those found in other prevalence studies [30,31]. In contrast with what is currently known [12,32,33], no correlation was found between prolonged QTc and the gender or age of the patients. Furthermore, despite MTD’s capacity for storage in the body [8], no correlation was found between MTD treatment duration and QTc values. However, in accordance with recent research on the subject [33], a linear relationship was discovered between the patients’ MTD dose at the time the ECG was performed and the QTc value, implying that MTD dosage plays a role in the prolongation of the QTc interval.

No-one in the sample of patients with prolonged QTc was taking any drugs that had a known risk of QTc prolongation, and only a minority of them were taking drugs that had a possible or conditional risk of QTc prolongation and TdP. It can be deduced that in the vast majority of patients, the prolongation of the QTc interval was caused by MTD alone, and that in the remaining patients, MTD may have played a synergistic role with the other drugs.

Another point to note is that benzodiazepines were used by more than a third of the patients in the study; even though they play no role in QTc interval prolongation, benzodiazepines should be used with caution in opioid-addicted patients being treated with MTD because they too can be abused and can cause respiratory depression in conjunction with MTD [34,35].

Given the lower cardiotoxic ability of LevoMTD compared to the racemic mixture (which contains equal proportions of both enantiomers) usually administered to patients [36], we investigated the role of LevoMTD in the normalization of QTc values in patients who had a prolongation of the QTc value following MTD treatment; the QTc value was completely normalized in 9 of the 10 patients who underwent the therapy change, confirming the potential benefits of LevoMTD maintenance therapy. Despite the small sample size, this study confirms the benefits of using LevoMTD in patients who have risk factors for QTc prolongation. Despite its higher cost [37], LevoMTD should be recommended as a first-line treatment for patients with pre-existing structural cardiac diseases, patients who take QTc-prolonging drugs, patients with other risk factors, or patients who already have a prolonged QTc interval prior to MTD treatment. In clinical practice, it is therefore necessary to thoroughly evaluate each patient in order to determine individualized therapies [38]. Furthermore, if other studies confirm the correspondence between manual and automatic QTc measurements, the evaluation of patients will become easier for clinicians with fewer cardiologic skills, as in low-access threshold services and/or risk-reduction programs. Even though there is plenty of research on MTD’s propensity to lengthen the QTc interval, very few studies have investigated the impact of levoMTD in correcting this risky side-effect.

One of the study’s limitations is the absence of a baseline ECG prior to the start of MTD treatment. Another limitation is the lack of information on the patients’ plasmatic electrolyte levels; in particular, hypokalemia can be associated with QTc elongation [39] and should thus be monitored. Other limitations that should be mentioned are the lack of a thorough history of the patients, the possibility of the patients abusing other illicit drugs, and the variability in their adherence to therapy.

## 5. Conclusions

This study showed a linear relationship between the daily dose of MTD and the measured QTc value; however, there appeared to be no relationship between the QTc value and the length of the therapy.

It was observed that one-eighth of the patients taken into consideration had a QTc prolongation over the normal ranges, validating the inclusion of MTD among the lists of QTc-prolonging drugs.

LevoMTD’s weaker association with QTc prolongation and TdP has been confirmed by the observation that 90% of patients receiving MTD treatment with a QTc prolongation underwent a normalization of the QTc value (<470 ms in men and <480 ms in women) after switching from MTD to levoMTD.

In order to reduce the risk of QTc prolongation and the mortality connected with it, this study has demonstrated the recommendation of an accurate, familiar, and pathological anamnesis in patients needing MTD, the execution of a basal ECG prior to the start of the treatment, and a thorough evaluation of other risk factors (such as serum levels of electrolytes and concomitant therapies). Furthermore, as LevoMTD is a safer drug, it should be made available in all addiction services.

## Figures and Tables

**Figure 1 biomedicines-11-02109-f001:**
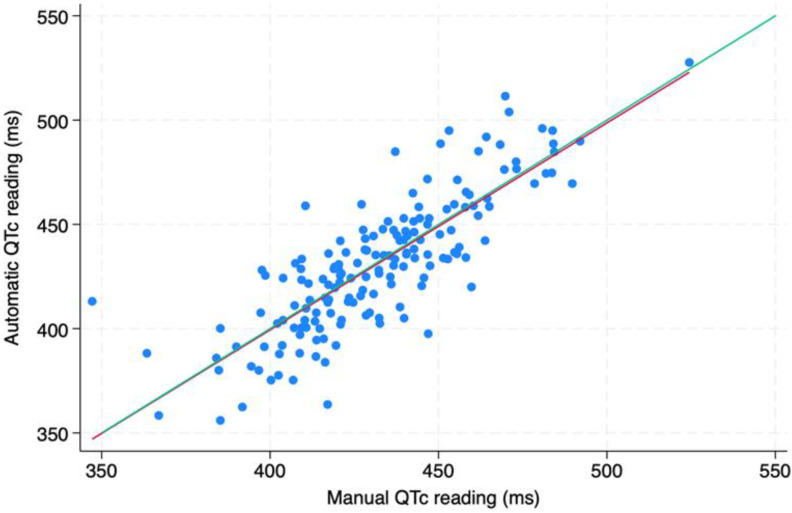
Comparison of manual versus automatic QTc reading. The regression line (in red) is superimposed to the identity line (in green).

**Figure 2 biomedicines-11-02109-f002:**
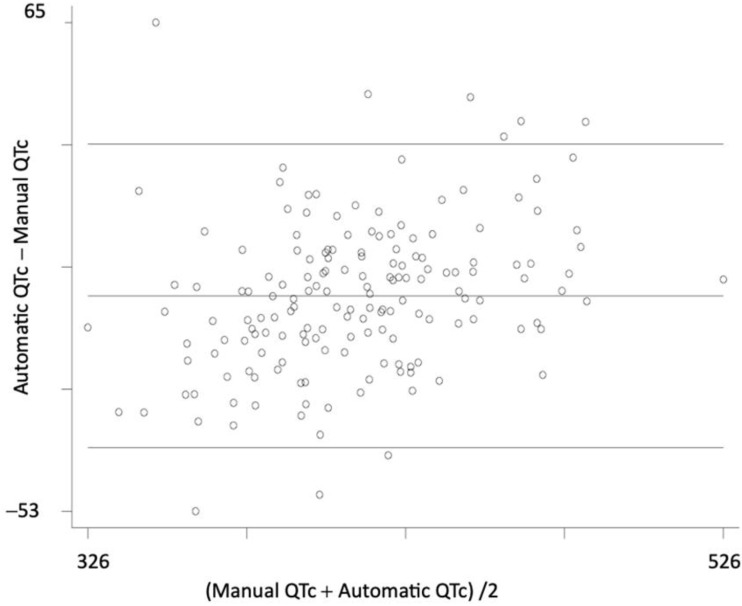
Plot according to Bland–Altman, where the mean of the two measurements is reported on the X axis and the difference on the Y axis.

**Figure 3 biomedicines-11-02109-f003:**
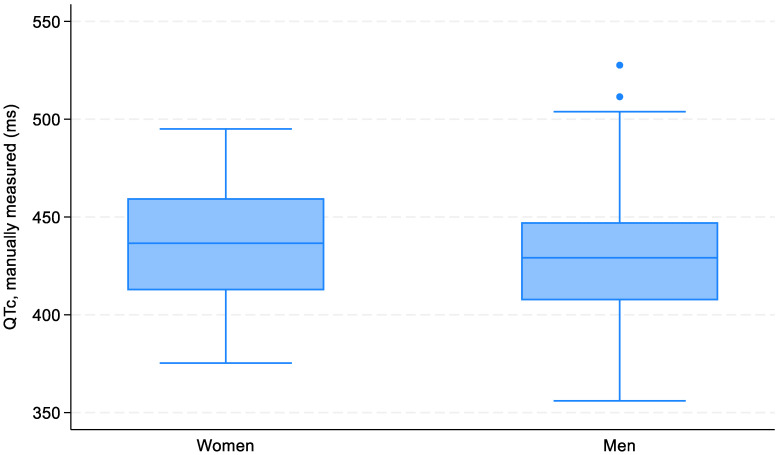
Box-and-whiskers plot of the QTc distribution based on sex.

**Figure 4 biomedicines-11-02109-f004:**
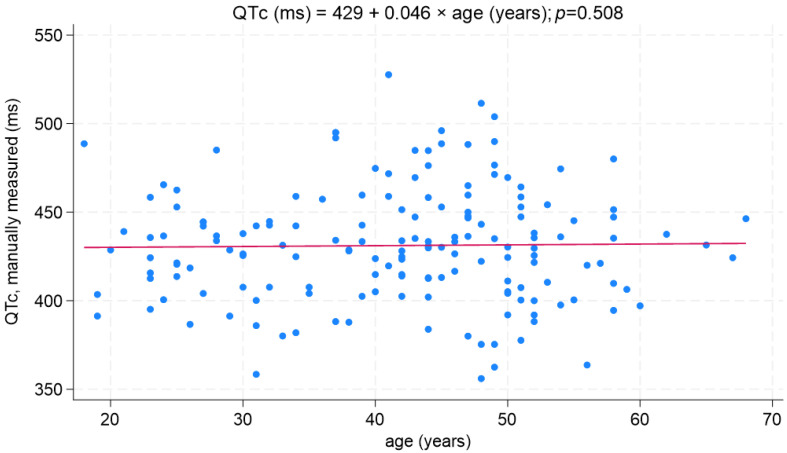
QTc distribution as a function of age.

**Figure 5 biomedicines-11-02109-f005:**
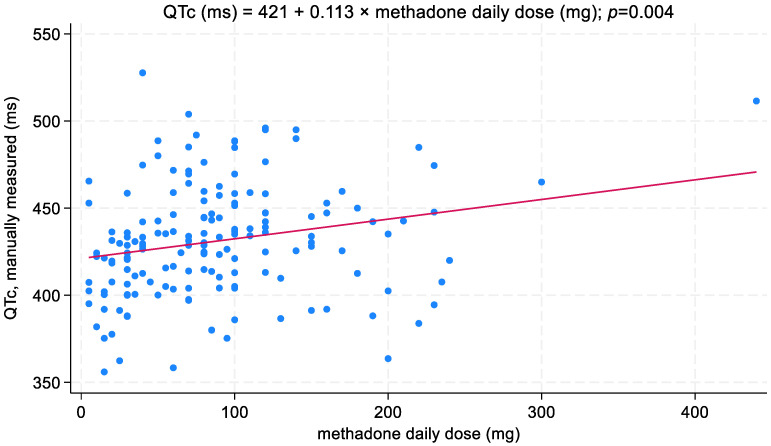
Relationship between QTc interval and daily dosage of MTD.

**Figure 6 biomedicines-11-02109-f006:**
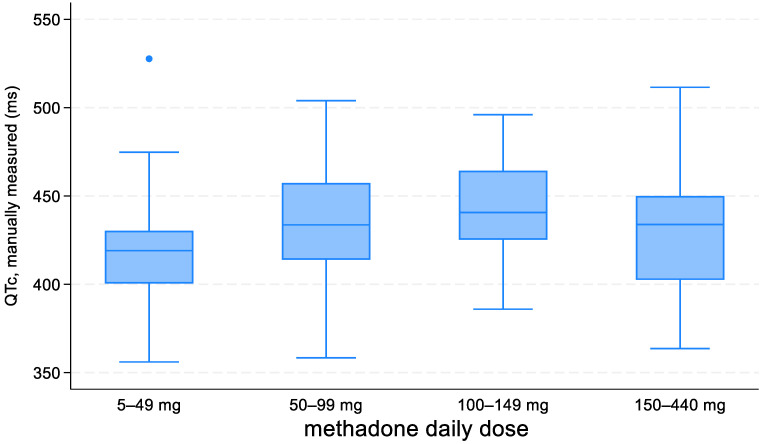
QTc interval as a function of methadone daily dose, coded over four levels.

**Figure 7 biomedicines-11-02109-f007:**
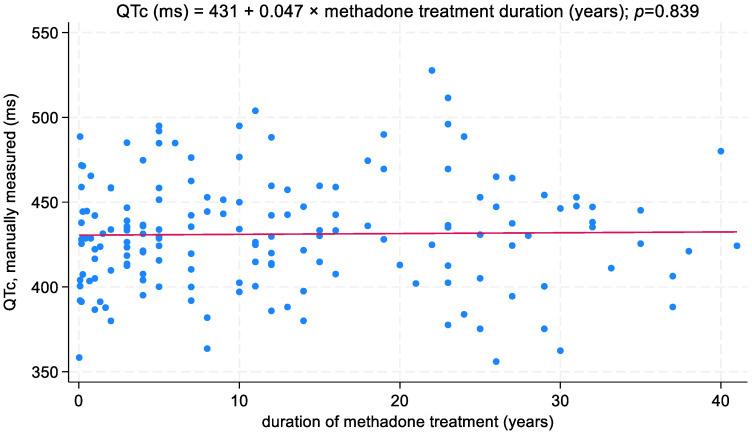
Relationship between QTc interval and duration of MTD treatment.

**Figure 8 biomedicines-11-02109-f008:**
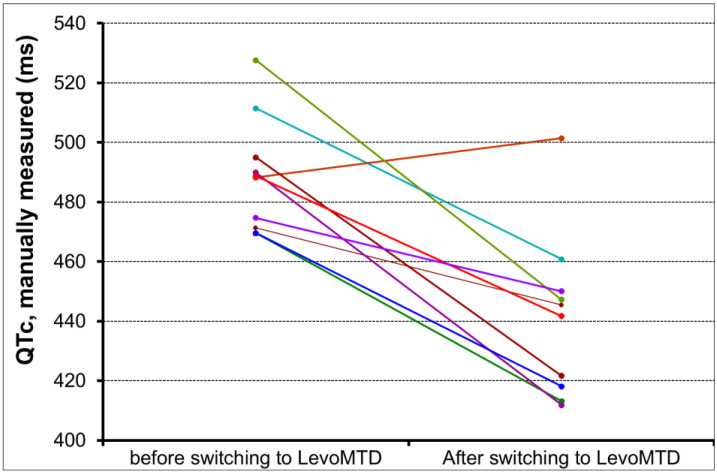
Comparison between QTc values before and after switching to LevoMTD. Each colored lined refers to a single subject.

**Table 1 biomedicines-11-02109-t001:** Drugs with a known risk of TdP [13].

Antiarrhythmic drugs	Disopyramide, Dofetilide, Dronedarone, Flecainide, Hydroquinidine, Ibutilide, Nifekalant, Procainamide, Quinidine, Sotalol
Antibiotics	Azithromycin, Ciprofloxacin, Clarithromycin, Erythromycin, Moxifloxacin, Levofloxacin
Antidepressants	Citalopram, Escitalopram
Antiemetic drugs	Domperidone, Ondansetron
Antifungal drugs	Fluconazole, Pentamidine
Antimalarial drugs	Chloroquine, Hydroxychloroquine
Antineoplastic drugs	Aclarubicin, Mobocertinib, Oxaliplatin, Vandatenib
Atypical Antipsychotics	Sulpiride
General Anesthetics	Propofol, Sevoflurane
Opioid Agonists	MTD
Typical Antipsychotics	Haloperidol, Chlorpromazine, Chlorprothixene, Droperidol, Levomepromazine, Levosulpiride, Thioridazine, Pimozide
Others	Anagrelide, Bepridil, Cilostazol, Cocaine, Terlipressin

**Table 2 biomedicines-11-02109-t002:** Determinants of the QTc interval, evaluated by multiple linear regression. Coefficients represent the estimates of QTc prolongation in msec.

	Coefficient in ms (95% Confidence Interval)	*p* Value
Dose of MTD (mg/day)		
5–49	1 (reference)	---
50–99	18.0 (5.4, 30.5)	0.005
100–149	28.8 (14.8, 42.8)	<0.001
150–440	15.2 (−0.3, 30.7)	0.054
Duration (years)		
0 (0–9)	1 (reference)	---
1 (10–19)	1.5 (−11.5, 14.5)	0.824
2 (≥20)	−1.0 (−15.0, 13.1)	0.893
Gender (men vs. women)	−8.4 (−22.0, 5.1)	0.221
Age (per 10 years increase)	1.2 (−4.2, 6.6)	0.656
Antidepressants (Yes vs. No)	0.7 (−12.5, 14.0)	0.915
Typical antipsychotic drugs (Yes vs. No)	−2.5 (−24.4, 19.4)	0.823
Atypical antipsychotic drugs (Yes vs. No)	−4.4 (−18.8, 10.1)	0.550

**Table 3 biomedicines-11-02109-t003:** Characteristics of patients with QTc elongation.

Variable	*n*	Mean	SD	Min–Max
QTc (ms)	23	487	14	470–528
Age (years)	23	43	8.5	18–58
Treatment duration	23	12 years	10 years	1 month–40 years
Daily dosage (mg)	23	113	87	40–440

**Table 4 biomedicines-11-02109-t004:** QTc variation following therapy change from MTD to levoMTD.

	Sex	QTc with MTD (ms)	QTc with LevoMTD (ms)	Difference (ms)
1	M	511	461	50
2	M	488	501	−13
3	M	470	413	57
4	F	495	422	73
5	M	528	447	81
6	M	471	445	26
7	M	490	412	78
8	M	470	418	52
9	M	489	442	47
10	M	475	450	25

## Data Availability

Data supporting reported results can be found by contacting medicina.dipendenze@aovr.veneto.it.s.

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
