# Peer review of "Methadone Maintenance and QT-Interval: Prevalence and Risk Factors—Is It Effective to Switch Therapy to Levomethadone?"

_biomedicines, 2023, doi:10.3390/biomedicines11082109_

Round 1

Reviewer 1 Report

Review on the manuscript of Santin, L. et al.,: “Methadone maintenance and QT-interval: prevalence and risk factors. Is it effective to switch therapy to Levomethadone?”.

In this manuscript, the authors explored the influence of methadone on the QTc interval and how Levomethadone could be beneficial to this physiological parameter. In 165 patients using methadone (distinct dosages and periods of use), authors describe a linear association between the dosage of methadone and QTc length; however, no correlation between QTc value and age, gender or duration of therapy was found. In 90 % of the patients with QTc prolongation (9/10), the change of their therapy from methadone to Levomethadone normalized the QTc length.

The manuscript is very clear and well written. However, some issues arise to me in its current format. So, I hope the authors find the following comments and suggestions useful.

1 - For the groups of patients that switched therapy from methadone to Levomethadone (10 patients), 1 of them registered an increase in the QTc interval. Authors refer that this patient was under Levomethadone for the shortest period of time (1 month and 20 days). For the other patients who registered a decrease in QTc length after switching to Levomethadone, is there any relationship between the time under Levomethadone treatment and the decrease in the QTc interval? If it does, it supports the authors’ hypothesis.

2 - For the groups of patients that switched therapy from methadone to Levomethadone (10 patients), the 9 patients that registered a decrease QTc interval (when compared to the QTc interval registered under methadone treatment) where the ones who were taking other medicines that could interfere with the QTc interval? Or those patients (that were using medicines that could prolong the QTc interval) did not register a prolongation of QTc interval? Can authors make this point clear in the manuscript?

Reviewer 2 Report

This study investigated the prevealnce and potential risk factors for QTc intreval prlongation in a patient population under maintanance therapy with methadone as well as the effects of the switch to levomethadone in some cases with significant QTc prolongation. The topic is of interest and of clinical relevance, however, some additions, explanations and a different presentation of some of the findings would improve the readibility and quality of the manuscript.

Somer comments and suggestions:

Abstract:

-       Currently some number are included ((1), (2) etc.), is that necessary? Perhaps better to include subsections instead, e.g. “background”, “methods”, “results”, “conclusion”

-       The design of the study should be added here and in the methods section (prospective/retrospective?)

-       Better to use past tense here and in other partsof the manuscript, e.g. “The study of this purpose was…”

-       Add how a prolongation of the QTc interval was defined in this study

Introduction:

-       Some references to be added after some sentences, e.g. “Methadone (MTD) is the most commonly used drug in the world to treat opioid ad-31 diction due to its safety, efficacy, and low cost.

-       Some language editing might be needed to avoid strange phrasing such as “the need to commit cromes”, also not sure for example what “stypsis” is

-       You mention “MTD is a chiral compound and its two enantiomers (R and S) have different pharmacokinetic and pharmacodynamic properties” but only pharmacodynamic differences are discuss in the manuscript. Could you also shortly mention their pharmacokinetic differences?

-       Table 1: Reference to be added, also some risk factors appear to be missing, for example hypomagnesemia

-       Table 2: Reference to be added, also I assume that this is only a selection of drugs with a known risk, in this case this should be clarified

Methods:

-       You used a different test to test for statistically significant differences of the QT before and after switch to levomethadone, this should be added

-       Also add the statistics software you used

-       Later in the results some findings are presented as mean (SD), some as median (range) and some with combinations of those too. It would be better to clarify in the methods how you are going to present your data (usually mean (SD) if normally distributed and median (range) if not normally distributed) and then consequently do this in the results

-       Because of the different potency, a decrease of the dose is indicated when switching from methadone to levomethaodne. Was it done like this in this study? Needs to be described in the methods and/or the results

-       Did an ethics committee apporve this study? Did the participants provide consent? This informatiopn should be added

Results

-       Currently too many tables, which often present data already mentioned in the text. Try to summarizes your findings in less tables and also either mention them in the text or in the table, not both

-       Normally and not normally distribited data to be presented the way you are going to describe this in your methods

-       Figure 1: Which axis is which? To be added. Also for some it seems that the two measurements were quite different, even if this is not statistically significant when you look at the whole sample. Did you consider performing the same analyses with the automatic QTc readings to see if some of the results would be different?

-       Figure 2: Same here, not clear what each axis is

-       Figures showing correlations: Add the correlation line in the graph, also the information in the text if this was a strong correlation or not

-       Figure 7: What is the x-axis here?

-       Section 3.5: What was the reason for not switching in the other 13 cases? How was the dose adjusted when switching from methadone to levomethadone? What was the reason for waiting that long in some cases (up to 48 months, that is a very long time) to perform the control ECG after switching?

-       Table 11: Could you also calculate the differences and add this information in a separate column?

Discussion:

-       The two first paragraphs are more part of/repetition of the introduction. Better to start your discussion by qualitativley summarize the findings of your study (this is what you currently do in your 3rd paragraph in the discussion)

-       Somewheer in the discussion you need to compare your finding with other similar studies done previously/what is already known in this field (for example, that the QTc prolongation depend on the dosage is already known, focus on what your study adds to the current knowledge)

-       In general, avoid repetitions between the discussion and the introduction, they shouldn’t include the same information twice

-       “It is also worth noting that the only patient who did not have a QT value normalization was the one who had more recently switched”: In this case the switch took place almost 2 months before the control ECG. How long do you think it would take for the QTc to normalize/decrease?

Conclusion:

-       Same comment with the discussion, avoid repeating information, make it shorter, focus on the summary of the findings of your study and their implications for clinical practice

Thank you for letting me review this work. For specific comments and suggestions please s. comments to the authors.

Round 2

Reviewer 2 Report

Thank you for addressing all the comments.

Some very minor editing might be of advantage.

Author Response

The final version of the manuscript has been revised by a native speaker.